# Neoadjuvant or concurrent atezolizumab with chemoradiation for locally advanced cervical cancer: a randomized phase I trial

Jyoti Mayadev [1,18] ✉, Dmitriy Zamarin [2,18], Wei Deng[3], Heather A. Lankes [4], Giulio Pesci[5], Hayeon Kim[6], Junzo P. Chino[7], Barbara Banbury [8], Ned Sherry[8], Elad Sharon[9], Sharad A. Ghamande[10], Catherine Ferguson[10], Loren Mell[1], Laura Holman[11], Cara Mathews [12], David O'Malley [13], Alexander Olawaiye[14], Elizabeth Hopp[15], Charles Leath III[16], Larry Copeland[13], Robert Mannel[11], Roisin O'Cearbhaill [5], Carol Aghajanian[5] & Russell J. Schilder[17]

Combined immune checkpoint blockade (ICB) and chemoradiation (CRT) is approved in patients with locally advanced cervical cancer (LACC) but optimal sequencing of CRT and ICB is unknown. NRG-GY017 (NCT03738228) was a randomized phase I trial of atezolizumab (anti-PD-L1) neoadjuvant and concurrent with CRT (Arm A) vs. concurrent with CRT (Arm B) in patients with high-risk node-positive LACC. The primary endpoint was the fraction of expanded tumor-associated T-cell receptor (TCR) clones in blood at day 21 as a surrogate measure of anti-tumor immune response. Secondary objectives were safety and feasibility, 2-year disease-free survival (DFS), and predictive value of PD-L1 expression. Forty patients were randomized, 36 received treatment, and 25 were evaluable for the primary endpoint. After cycle 1, there was peripheral expansion of higher proportion of tumor-associated TCR clones in Arm A than in Arm B ($p = 0.0025$) that remained higher at day 21, meeting the pre-specified endpoint on two-sample T-test ($p = 0.052$), but not on sensitivity analysis by Wilcoxon test ($p = 0.13$). At the median follow up of 25.8 months, 2-year DFS was 76% in Arm A and 56% in Arm B ($p = 0.28$). There were no new safety signals. In conclusion, neoadjuvant ICB prior to CRT was safe and was associated with immunologically and clinically favorable outcomes, warranting larger confirmatory studies.

Patients with lymph node-positive locoregionally advanced cervical cancer (LACC), and in particular, patients with para-aortic lymph node (PALN) metastases treated with chemoradiation (CRT) have a dismal 3-year overall survival (OS) and progression-free survival (PFS) of 39% and 34%, respectively[1]. There is a critical need to develop more effective therapeutic strategies and novel methods to identify patients at high risk of relapse.

Immunotherapy including checkpoint blockade (ICB) have demonstrated activity in patients with recurrent/metastatic cervical cancer[2–5]. There are limited clinical data surrounding the optimal sequencing needed to provoke an ideal immune response when combining immunotherapy with RT, while biomarkers of response remain unknown[6]. Evaluation of T-cell receptor (TCR) repertoire in tumors and peripheral blood has emerged as an important tool in the evaluation of

**Fig. 1 | Consort diagram and study schema. A** Consort diagram of the randomized trial participants and treatment disposition. **B** Study schema with the Arm A and Arm B treatment, imaging, and biospecimen collection timepoints. Created in BioRender. Zamarin, D. (2022) BioRender.com/g30o610.

immune response within the context of immunotherapy[7]. TCR sequencing data can be used to assess the clonal richness and diversity of lymphocyte populations; to track specific TCR clonotypes over time, between tissues, and across lymphocyte subsets; to detect clonal expansion; and to detect the recruitment of new clones into a tissue which may be correlated with therapeutic response[7].

Here, we report on the outcomes of NRG-GY017, a randomized phase I trial of the anti-PD-L1 antibody atezolizumab neoadjuvant and concurrent (Arm A) or concurrent with CRT (Arm B) in patients with node-positive LACC (ClinicalTrials.gov Identifier: NCT03738228). The primary objective was to evaluate the expansion of tumor-associated TCR clones in peripheral blood as a surrogate measure of immune response, with the secondary objectives of safety, tolerability, and 2-year disease-free survival (DFS). We demonstrate that neoadjuvant atezolizumab administration results in peripheral expansion of a higher proportion of tumor-associated TCR clones in Arm A than in Arm B and is associated with favorable 2-year DFS. No new safety signals were identified. These findings support further evaluation of the neoadjuvant IO and CRT sequencing strategy in LACC.

## Results

### Patients

Forty patients were enrolled and randomized as shown in Fig. 1A consort diagram. The characteristics of the 36 eligible and treated patients (19 in Arm A and 17 in Arm B) are summarized in Table 1. Median age was 48 years old, 22% were Hispanic or Latino, and 19% were Black or African American. Based on the pre-treatment PET/CT, 36% of patients had pelvic and para-aortic lymph nodes (PALN), 3% PALN, 47% pelvic LN, 14% not otherwise specified. While the arms were mostly well-balanced, Arm A (neoadjuvant) enrolled patients that were likely to be older ($p = 0.005$), Hispanic or Latino ($p = 0.03$), had lower PD-L1 tumor cell positivity ($p = 0.023$), and had numerically more patients with PALN positivity (not statistically significant).

### Safety

Secondary outcomes of treatment-related grade 3 or higher adverse events are summarized in Table 2. Arm A had 16 DLT-evaluable patients out of the 19 eligible and treated, and 0% DLTs; Arm B had 14 DLT-evaluable patients out of the 17 eligible and treated, and 3 patients had DLTs. Of the three DLTs, one was a grade 3 immune-related colitis, likely not RT related as the patient did not respond to anti-diarrhea medication and did respond to steroids; one had a grade ≥ 3 colitis (non-immune related, likely RT), and one had thrombocytopenia causing a cisplatin delay of > 2 weeks.

Supplementary Table 1 summarizes the distribution of patients for the secondary outcomes by the highest grade of adverse related to treatment. One grade 5 myocardial infarction observed in Arm A was unrelated to treatment. Among the 36 patients, 75% completed their treatment; treatment was discontinued due to disease progression ($n = 1$), death ($n = 1$), toxicity ($n = 4$), other ($n = 1$), and withdrawal of consent ($n = 2$). Of all patients, 83% received 3 doses of atezolizumab, 86% received at least 4 doses of cisplatin, 94% (34/36) completed per protocol EBRT of 45 Gy with an external beam boost to 54–58 Gy to the involved lymph nodes depending on location, and 94% (34/36) received brachytherapy with a prescription dose of 6-7 Gy in 4-5 fractions (please see radiation section of the protocol in the appendix for further radiation prescription details). Of the 2 patients who did not complete RT: one had death unrelated to therapy, and one withdrew consent.

### Efficacy

The secondary outcomes of efficacy were recorded and shown in Table 3. There were 3 patients whose 2-year DFS was not evaluable: two due to withdrawal of consent and one due to loss of follow-up. At the median follow-up of 25.8 months, the proportion of patients that achieved 2-year DFS was 76% in Arm A and 56% in Arm B ($p = 0.28$). The survival difference between the two arms was explored using permutation-based log rank test with a p-value of 0.339 for DFS (Fig. 2A) and a $p$-value of 0.425 for the OS (Supplementary Fig. 1). Both KM median estimates for DFS and OS were not reached; the KM estimates for 2-year DFS were 78% in Arm A and 57% in Arm B (Fig. 2A), and the KM estimates for 2-year OS were 78% in Arm A and 87% in Arm B (Supplementary Fig. 1). The presence of PALN was associated with shorter DFS and OS, albeit number of events was small, and 5 patients had unknown PALN status (Supplementary Fig. 2). In an exploratory analysis of the day 28 biopsy, 11/16 in Arm A (69%) and 6/15 in Arm B (40%) had evidence of pathologic treatment effect, characterized as complete or partial regression of tumor cells $p = 0.1556$ (Fig. 2B). Association between pathologic treatment effect and 2-year DFS was evaluated by chi-square test with a $p$-value of 0.0132. The secondary outcome association between the baseline PD-L1 tumor and immune cell scoring and 2-year DFS was explored using Spearman's rank correlation coefficient and no statistically significant correlation between the baseline PD-L1 status and DFS was found ($p = 0.178$ for tumor scoring, $p = 0.372$ for immune cell scoring). Although the increase in PD-L1 staining was observed in response to therapy in some patients, this was not uniform across all patients and did not meet statistical significance (Supplementary Fig. 3). It should be noted that the trial

## Table 1 | Patient Demographics and Characteristics

| | Arm A (n = 19) | Arm B (n = 17) | Total (n = 36) | p-value[a] |
|---|---|---|---|---|
| Age (median, min-max) | 56 (35–71) | 43 (24–60) | 47.5 (24–71) | 0.005 |
| Ethnicity | – | – | – | 0.028 |
| Hispanic or Latino | 7 (36.8%) | 1 (5.9%) | 8 (22.2%) | – |
| Not Hispanic or Latino | 11 (57.9%) | 16 (94.1%) | 27 (75.0%) | – |
| Not Reported | 1 (5.3%) | 0 (0.0%) | 1 (2.8%) | – |
| Race | – | – | – | 0.597 |
| Black or African American | 3 (15.8%) | 4 (23.5%) | 7 (19.4%) | – |
| White | 14 (73.7%) | 13 (76.5%) | 27 (75.0%) | – |
| Not Reported | 2 (10.5%) | 0 (0.0%) | 2 (5.6%) | – |
| Performance status | – | – | – | 0.717 |
| 0 | 13 (68.4%) | 13 (76.5%) | 26 (72.2%) | – |
| 1 | 6 (31.6%) | 4 (23.5%) | 10 (27.8%) | – |
| Histology | – | – | – | 0.493 |
| Adenocarcinoma NOS | 4 (21.1%) | 1 (5.9%) | 5 (13.9%) | – |
| Adenosquamous | 1 (5.3%) | 2 (11.8%) | 3 (8.3%) | – |
| Squamous Cell Carcinoma | 14 (73.7%) | 14 (82.4%) | 28 (77.8%) | – |
| FIGO stage | – | – | – | 0.988 |
| IB | 3 (15.8%) | 3 (17.6%) | 6 (16.7%) | – |
| IIB | 12 (63.2%) | 10 (58.8%) | 22 (61.1%) | – |
| IIIB | 3 (15.8%) | 4 (23.5%) | 7 (19.4%) | – |
| IVA | 1 (5.3%) | 0 (0.0%) | 1 (2.8%) | – |
| Baseline PET/CT SUV max for cervix (median, min -max) | 18.85 (5.1–36) | 16.5 (8.4–35.9) | 18.3 (5.1–36) | 0.75 |
| Missing | 1 | 0 | 1 | – |
| Para-aortic lymph node metastases (PET/CT) | – | – | – | 0.482 |
| No | 7 (36.8%) | 10 (58.8%) | 17 (47.2%) | – |
| Yes | 9 (47.4%) | 5 (29.4%) | 14 (38.9%) | – |
| Not available | 3 (15.8%) | 2 (11.8%) | 5 (13.9%) | – |
| Pre-treatment PD-L1 (SP263) immune score | – | – | – | 0.588 |
| Median (min – max) | 3 (0–40) | 1 (0.5–5) | 2.5 (0–40) | – |
| Negative ( <1%) | 2 (10.5%) | 3 (17.6%) | 5 (13.9%) | – |
| Positive ( ≥1%) | 8 (42.1%) | 9 (52.9%) | 17 (47.2%) | – |
| missing | 9 (47.4%) | 5 (29.4%) | 14 (38.9%) | – |
| Pre-treatment PD-L1 (SP263) tumor cell score | – | – | – | 0.023 |
| Median (min – max) | 0.5 (0–95) | 12.5 (0.5–100) | 1 (0–100) | – |
| Negative ( <1%) | 7 (36.8%) | 2 (11.8%) | 9 (25.0%) | – |
| Positive ( ≥1%) | 3 (15.8%) | 10 (58.8%) | 13 (36.1%) | – |
| missing | 9 (47.4%) | 5 (29.4%) | 14 (38.9%) | – |

[a]observations with missing values were included for discrete-type variables and excluded for continuous variables.

## Table 2 | Grade 3 or Higher Treatment-Related Adverse Events

| AE type | Arm A (n = 19) | Arm B (n = 17) |
|---|---|---|
| | No. and (%) of Patients Grade 3–5 | No. and (%) of Patients Grade 3–5 |
| Overall Highest Grade | 3 (15.8) | 10 (58.8) |
| Anemia | 0 (0.0) | 2 (11.8) |
| Febrile neutropenia | 0 (0.0) | 2 (11.8) |
| Diarrhea | 1 (5.3) | 3 (17.6) |
| Mucositis oral | 0 (0.0) | 1 (5.9) |
| Nausea | 0 (0.0) | 2 (11.8) |
| Vomiting | 0 (0.0) | 2 (11.8) |
| Urinary tract infection | 1 (5.3) | 1 (5.9) |
| Lymphocyte count decreased | 0 (0.0) | 2 (11.8) |
| Neutrophil count decreased | 1 (5.3) | 2 (11.8) |
| Platelet count decreased | 0 (0.0) | 1 (5.9) |
| White blood cell decreased | 0 (0.0) | 1 (5.9) |
| Hypocalcemia | 0 (0.0) | 1 (5.9) |
| Hypokalemia | 0 (0.0) | 1 (5.9) |
| Hypomagnesemia | 0 (0.0) | 2 (11.8) |
| Chronic kidney disease | 1 (5.3) | 0 (0.0) |

was not powered to detect significant differences in its secondary survival endpoints between the study Arms.

### Evolution of TCR repertoire in response to therapy

Among the 36 eligible and treated patients, there were 30 (17 on Arm A and 13 on Arm B) with pre- and on-treatment peripheral TCR sequencing data. Twenty-five patients (14 in Arm A and 11 in Arm B) had additional tumor TCR sequencing data available, which enabled tracking of tumor-associated TCR clones in peripheral blood. For the primary outcome, at day 21, there was a significant decrease in overall TCR diversity (reflecting the number of unique rearrangements) from baseline in both arms in response to treatment ($p = 0.0004$ for Arm A and $p = 0.0002$ for Arm B) with majority of the loss seen in the CRT phase (Supplementary Fig. 4a). Notably, this was accompanied by the

decrease in T cell fraction (estimated by TCR sequencing as the total number of T cells out of the total number of nucleated cells in the sample) and overall leukopenia observed during the CRT phase (Supplementary Fig. 4b). Per pre-specified exploratory objectives, evolution of overall TCR repertoire and tumor-associated TCR repertoire was then examined (Fig. 3A, B). Overall expansion and appearance of new TCR clones (clonal expansion) from baseline to day 21 was observed in both Arm A ($p < 0.0001$) and Arm B ($p = 0.0002$), though the difference between the arms was not significant ($p = 0.36$) by 2-sided $t$ test (Fig. 3C). Most of the clonal expansion occurred during CRT as evidenced by the clonal expansion after the first cycle in each arm, with significantly higher increase observed in Arm B vs. Arm A (median 132 vs. 76.5, respectively, $p = 0.017$) (Fig. 3D). Since overall clonal expansion could be predominantly indicative of a non-specific inflammatory response to radiation, we evaluated the relative expansion of tumor-associated T cell clones out of total expanded clones as a surrogate measure of T cell repertoire that is presumed to be enriched for tumor-reactive T cells (Fig. 3A)[8]. After cycle 1 of therapy, significantly higher proportion of peripherally expanded clones were tumor-associated in Arm A vs. Arm B (mean 0.37 vs. 0.09, $p = 0.0025$ by a Wilcoxon rank sum test) (Fig. 3E). At 10% significance level, at the day 21 primary objective endpoint, the peripheral proportion of tumor-associated TCR clones in Arm A remained higher than in Arm B ($p = 0.052$ by protocol-specified 2-sample $t$ test); however, this did not meet criteria for significance on sensitivity analysis using Wilcoxon rank sum test ($p = 0.13$) (Fig. 3F). Notably, contraction of the initially expanded tumor-associated clones in Arm A was observed during the CRT phase ($p = 0.048$) (Fig. 3G).

In patients evaluable for the primary objective, the associations between the fraction of tumor-associated TCR clonal expansion at day 21 and pre-treatment PD-L1 positivity in immune cells and in tumor cells were explored by Spearman rank correlation coefficient tests. Overall, at a 10% significance level, the results did not support the association between the fraction of TCR tumor-associated clone expansion at day 21 from baseline and pre-treatment PD-L1 in immune cells ($p = 0.849$) or tumor cells ($p = 0.3401$), although these results should be interpreted with caution since only 18 patients had available PD-L1 staining in this cohort.

Interestingly, when examining an exploratory endpoint the association between TCR repertoires and outcomes, at a 10% significance

**Table 3 | Clinical Outcomes for All Eligible and Treated Patients**

| | Arm A (n = 19) | Arm B (n = 17) | Total (n = 36) |
|---|---|---|---|
| Survival status | | | |
| Dead - Disease-related | 4 (21.1%) | 2 (11.8%) | 6 (16.7%) |
| Alive - Without Recurrence | 15 (78.9%) | 10 (58.8%) | 25 (69.4%) |
| Alive - With Recurrence | 0 (0.0%) | 5 (29.4%) | 5 (13.9%) |
| Number of doses of Atezolizumab | | | |
| 1 | 1 (5.3%) | 3 (17.6%) | 4 (11.1%) |
| 2 | 1 (5.3%) | 1 (5.9%) | 2 (5.6%) |
| 3 | 17 (89.5%) | 13 (76.5%) | 30 (83.3%) |
| Number of doses of cisplatin | | | |
| 0 | 1 (5.3%) | 0 (0.0%) | 1 (2.8%) |
| 2 | 1 (5.3%) | 1 (5.9%) | 2 (5.6%) |
| 3 | 1 (5.3%) | 1 (5.9%) | 2 (5.6%) |
| 4 | 2 (10.5%) | 2 (11.8%) | 4 (11.1%) |
| 5 | 3 (15.8%) | 6 (35.3%) | 9 (25.0%) |
| 6 | 11 (57.9%) | 7 (41.2%) | 18 (50.0%) |
| Dose modification | | | |
| No | 13 (68.4%) | 8 (47.1%) | 21 (58.3%) |
| Yes with at least 1 dose modification | 6 (31.6%) | 9 (52.9%) | 15 (41.7%) |
| RT Completion | | | |
| External Beam Dose per Protocol | 17 (89%) | 17 (100%) | 34 (94%) |
| Brachytherapy Treatment | 17 (89%) | 17 (100%) | 34 (94%) |
| Pathologic treatment effect on day 28 biopsy | | | |
| no tumor cells present, pathologic complete response | 5 (26.3%) | 5 (29.4%) | 10 (27.8%) |
| tumor cells present with treatment effect | 6 (31.6%) | 1 (5.9%) | 7 (19.4%) |
| tumor cells present | 5 (26.3%) | 9 (52.9%) | 14 (38.9%) |
| biopsy not done | 3 (15.8%) | 2 (11.8%) | 5 (13.9%) |
| Metabolic response at 3 months post-treatment based on PET/CT SUV max for cervix | | | |
| complete metabolic response | 5 (26.3%) | 6 (35.3%) | 11 (30.6%) |
| partial metabolic response | 0 (0.0%) | 6 (35.3%) | 6 (16.7%) |
| stable metabolic response | 2 (10.5%) | 0 (0.0%) | 2 (5.6%) |
| missing | 12 (63.2%) | 5 (29.4%) | 17 (47.2%) |
| Disease-free at 12 months | | | |
| No | 4 (21.1%) | 4 (23.5%) | 8 (22.2%) |
| Not evaluable | 1 (5.3%) | 1 (5.9%) | 2 (5.6%) |
| Yes | 14 (73.7%) | 12 (70.6%) | 26 (72.2%) |
| Disease-free at 24 months (2-year DFS) | | | |
| No | 4 (21.1%) | 7 (41.2%) | 11 (30.6%) |
| Not evaluable | 2 (10.5%) | 1 (5.9%) | 3 (8.3%) |
| Yes | 13 (68.4%) | 9 (52.9%) | 22 (61.1%) |

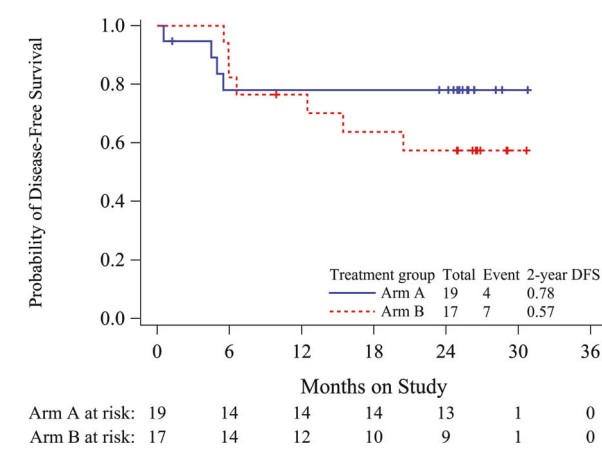

**A.**

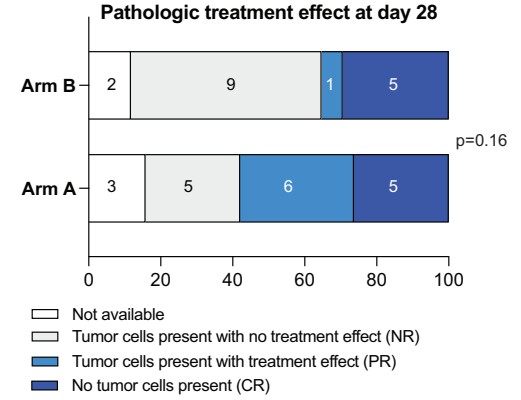

**B.**

**Fig. 2 | Clinical outcomes. A** Kaplan–Meier curve for disease-free survival by treatment. **B** Treatment effect in day-28 tumor tissue, 2-sided by Fisher exact test.

## Discussion

The optimal sequencing of RT and immunotherapy is not known. On one hand, administration of PD-1/PD-L1 blockade prior to RT may lead to re-invigoration of exhausted tumor-infiltrating T-cells, or prevention of exhaustion of newly infiltrating T-cells, with potentiation of radiation-mediated killing by T-cell response. On the other hand, pelvic RT carries the potential to kill activated proliferating T-cells, both intratumorally and in the tumor-draining lymph nodes. In fact, a recent study in preclinical cancer models suggested that irradiation of tumor-draining lymph nodes leads to decreased anti-tumor immune response and memory response[9]. Furthermore, T-cell exhaustion and senescence play a role in a dysfunctional T-cell response in virally driven cancers[10].

There are several clinical trials exploring the use of IO with CRT in LACC. A phase III randomized, double-blind trial that explored the use of durvalumab and CRT vs placebo and CRT followed by 24 months of durvalumab (CALLA) demonstrated no difference in the 2-year DFS with the addition of durvalumab to CRT[11]. However, KEYNOTE A18, a randomized double-blind clinical trial in LACC showed a 2-year PFS advantage with IO and CRT vs. CRT, 67.8% vs. 57.3%, HR = 0.70, p = 0.0020, resulting in the recent approval of pembrolizumab in combination with CRT for newly diagnosed patients with FIGO 2014 stage III and IV LACC[12]. The early results of a phase I trial examining the concurrent versus sequential sequence of pembrolizumab and concurrent CRT showed a safe toxicity profile[13]. These studies highlight a need for mechanistic understanding and exploration of differential sequencing of CRT and immunotherapy. Notably, the largest benefit in KEYNOTE A18 was observed in stage III-IVa disease, where most patients (90%) had PD-L1 positive cancer and where

level, DFS at 24 months was negatively associated with TCR clone expansion at day 21 (coefficient: 0.4889, p = 0.008) and TCR diversity change at day 21 from baseline (coefficient: 0.3965, p = 0.0362), respectively (Supplementary Fig. 5a, b). As highlighted above, since total clonal expansion is likely indicative of a nonspecific inflammatory response to radiation, we examined the association between 24-month DFS and tumor-associated TCR clonal expansion. As a post hoc endpoint, no statistically significant association between 24-month DFS and the fraction of expanded tumor-associated TCRs was observed in either arm, though this analysis was significantly underpowered due to the small number of events in this population (n = 3 for Arm A and n = 5 for Arm B) (Supplementary Fig. 5c).

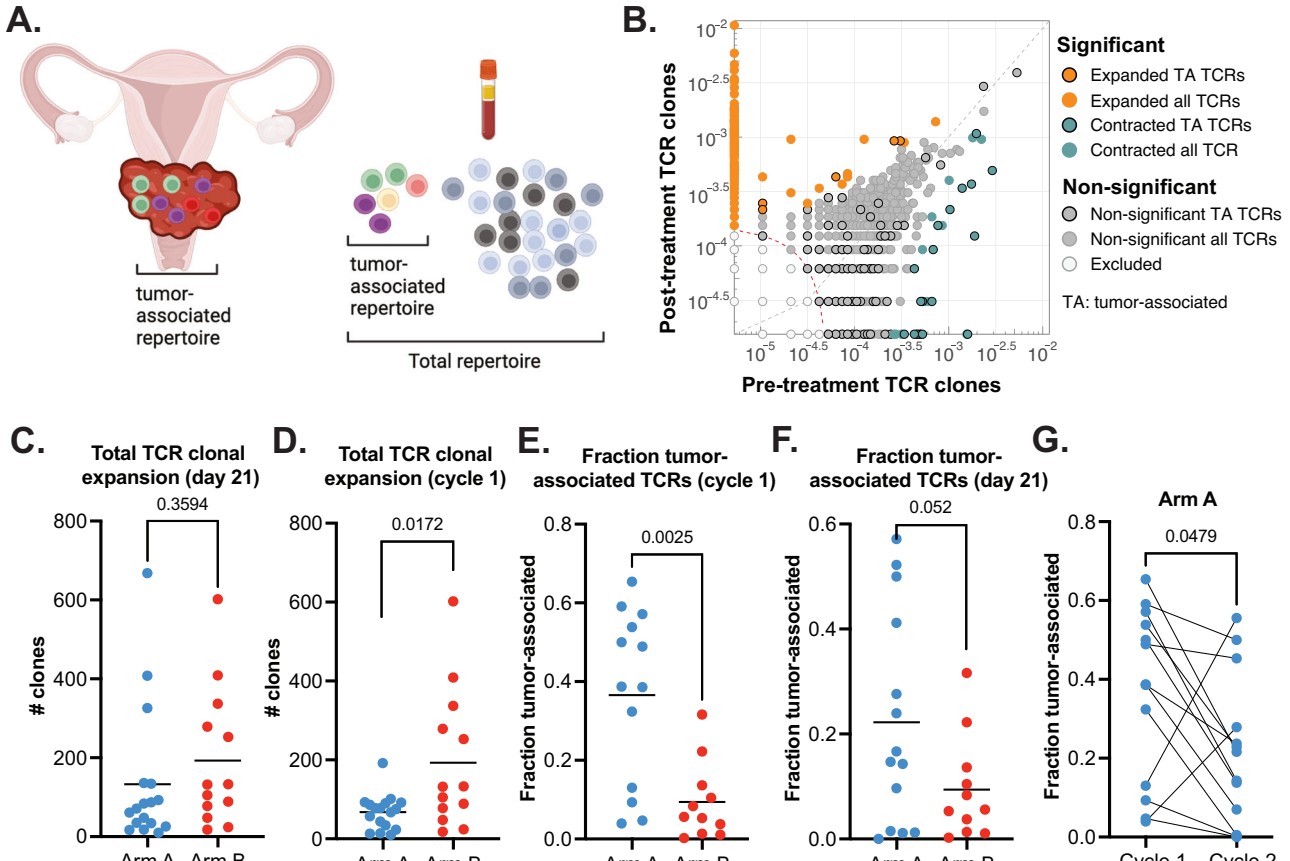

**Fig. 3 | Evolution of T cell repertoire parameters over the course of treatment.**
**A** Schematic of definitions of TCR repertoires in tumors (right) and peripheral blood (left), image created with BioRender. **B** Scatter plot from a representative patient demonstrating expansion and contraction from baseline to post-treatment. Each clone was tested for significant expansion using a binomial test after correcting for the control false discovery rate (FDR) at 0.01. **C** Total TCR clonal expansion at day 21. **D** Total TCR clonal expansion at completion of cycle 1. **E** Fraction of TCR clones that expanded at the end of cycle 1 of therapy and were found to be tumor-associated. **F** Fraction of TCR clones that expanded at day 21 and

were found to be tumor-associated. **C–E** Comparisons were performed using a two-sided Wilcoxon matched-pairs signed rank test. **F** Two-sided unpaired Students *T* tests were used for comparisons at a 10% significance level, pre-specified for the primary endpoint. **G** Change in tumor-associated TCR clonal percentages from cycle 1 to cycle 2 of therapy in Arm A (**C–D**) Arm A *n* = 17, Arm B *n* = 13. **E–F** Arm A *n* = 14, Arm B *n* = 11. **G** Arm A *n* = 13. Each data point represents an individual patient with an individual measurement for the specified time point. There were no adjustments for multiple comparisons. Source data are provided as a Source Data file. Image in 3 A created in BioRender. Zamarin, D. (2024) BioRender.com/u23c225.

limited benefit was observed for adenocarcinomas. Neoadjuvant sequencing, as seen in this trial could serve as a foundation for exploring the potential of neoadjuvant immunotherapy in PDL1-negative, locally advanced adenocarcinomas, for which treatment options are currently limited.

NRG-GY017 enrolled patients with high-risk clinical features: 86% had confirmed lymph node disease by PET/CT imaging and of those, 39% had PALN involvement. Notably, Arm A had potentially worse clinical and pathological features, including a higher proportion of adenocarcinomas, higher rates of PALN involvement, and lower tumor PDL-1 staining. Despite these adverse features, a promising signal of efficacy was seen in Arm A (2-year DFS of 76% in Arm A vs. 56% in Arm B), though the study was not powered for therapeutic efficacy comparison. These outcomes compare favorably to previous clinical trials of CRT[14,15] that included PALN-positive patients such as Retro EMBRACE and EMBRACE-1 studies, which had 41% and 52% of patients having lymph-node positive disease, with 5-year DFS ranging from 47%–76%[16,17]. Neoadjuvant and concurrent administration of atezolizumab with CRT was also associated with a higher rate of pathologic response on treatment, though not statistically significant. While pathologic response is not a validated endpoint in LACC, its potential association with 2-year DFS generates rationale for the evaluation of pathologic response as an early surrogate endpoint in future studies.

The use of TCR repertoire sequencing has enabled the evaluation of the evolution of immune response to immunotherapy and CRT. We find that there was a marked TCR clonal expansion in both arms, which was predominantly observed during the CRT phase. Despite this, only a minority of the peripherally expanding TCR clones were tumor-associated, suggesting that most of the expanding clones during CRT likely represent a non-specific inflammatory response. Supporting this, we found that overall clonal expansion was negatively associated with 2-year DFS (Supplementary Fig. 5). There was a significant loss of peripheral TCR diversity during CRT, suggesting that CRT may have deleterious consequences for immunogenicity. Indeed, patients with 2-year DFS exhibited a lower extent of TCR diversity loss in response to therapy (Supplementary Fig. 4). When specifically examining tumor-associated TCR clones, which are expected to be enriched for tumor-specific T-cells, we found that these clones represented a higher proportion of the peripherally expanding TCR clones in Arm A than in Arm B. Since the difference in the number of atezolizumab doses could potentially account for the difference in biological effects observed at day 21, we performed separate analyses comparing the biological effects after a single treatment cycle (day -21 to day 0 for Arm A and day 0 to day 21 for Arm B) (Fig. 3C). Notably these results indicate that most tumor-associated clonal expansion occurred in response to atezolizumab prior to initiation of CRT, suggesting that neoadjuvant

immunotherapy may mount an early systemic immune response that is sustained throughout CRT. Finally, while most of the diversity loss during the CRT was likely a nonspecific consequence of CRT-induced leukopenia, we also observed relative reduction of tumor-associated TCR clones in peripheral blood (out of total clones) in response to CRT, particularly in Arm A where these clones have previously expanded (Fig. 3G). This would suggest that radiation could indeed have direct negative impact on tumor-associated TCR clones, possibly through their direct killing in the tumor-draining lymph nodes.

This study illustrated high quality for therapy completion, RT oversight, and enrollment diversity. The addition of atezolizumab 3 weeks prior to CRT did not significantly change or delay CRT therapy, as 86% of patients completed 4- 6 cycles of concurrent cisplatin with CRT, 94% of patients completed EBRT and brachytherapy per protocol. Technologically advanced RT was employed with a workflow dedicated to high-quality assurance. Enrolling sites had to pass a credentialing process and submit pre-treatment RT rapid review for protocol compliance. The study enrolled a high percentage (37%) of underrepresented minorities.

The study has several limitations. The translational sample size was lower than targeted due to the study opening during the COVID-19 pandemic. While this expected attrition was pre-specified in the protocol, with power calculations allowing for attrition of up to 25% per arm to provide at least 80% power, the attrition was higher than expected when specifically focusing on the tumor-associated clone analyses. The small sample size could thus introduce bias and affect the generalizability of the findings. This trial was only open to select phase I NRG Oncology sites which may not be representative of all facilities. The patients had rigorous RT quality control and mostly had experienced providers with treatment at NCI-designated cancer centers, which has shown itself to improve outcomes in LACC despite controlling for the standard of care therapy[18]. Finally, the small sample size, although prespecified for attrition in the statistical methods section, may introduce bias and affect the generalizability of the findings.

In summary, NRG-GY017 is a hypothesis-generating clinical trial that explores sequencing of an immune-oncology agent and CRT with insight into the T-cell immune response with a favorable 2 year DFS in both arms. The trend toward superior pathological response, DFS, and tumor-associated T-cell expansion supports further evaluation of the neoadjuvant IO and CRT sequencing strategy in LACC.

## Methods

### Study population
This research complies with all relevant ethical regulations and has CIRB and institutional approval for the NRG Oncology network sites that participated (University of California San Diego IRB, Augusta University IRB, University of Oklahoma IRB, University of Pittsburg IRB, Ohio State University IRB, Women and Infants Brown University IRB, Medical College of Wisconsin IRB, University of Alabama IRB, Thomas Jefferson University IRB, Memorial Sloan Kettering IRB). Written informed consent was obtained by participants. The study, NRG GY017 (NCT03738228), was authorized and the study design complied with all relevant regulations regarding the use of human subject participants and was conducted in accordance with the criteria set by the Declaration of Helsinki. NRG-GY017 opened on 10/26/2018 and closed to accrual on 06/11/2020. The first patient was enrolled on January 7, 2019, and the last patient was enrolled on June 11, 2020.

Patients with confirmed LACC (squamous, adenocarcinoma, or adenosquamous) with FIGO 2009 stages IB2/IIA with positive PALN or IIB/IIIB/ IVA with positive pelvic and/or PALN were eligible.

### Treatment
NRG-GY017 is a two-arm, open-label randomized (1:1) prospective phase I trial with two experimental arms; Arm A: intravenous

atezolizumab 1200 mg every 3 weeks neoadjuvant and concurrent with CRT (days -21, 0, 21); Arm B: atezolizumab concurrent with CRT (days 0, 21, 42) (Design schema shown in Fig. 1) using block randomization from the NRG statistical office without blinding. Patients were enrolled in a consecutive manner with phase I NRG Oncology gynecology oncology sites with a safety monitoring rule dictating a study hold in the event of DLTs observed in more than 30% of DLT-evaluable patients in either one arm. Patients were treated with 6 weekly doses of cisplatin (40 mg/m²) concurrent with extended field radiation (EFRT) with Intensity Modulated Radiation Therapy (IMRT) to treat the pelvic and PALN and brachytherapy 80–90 Gy. All RT plans had a pre-treatment rapid review for RT quality. RT was to be completed within 56 days −/+ 3 days. Peripheral blood and tumor biopsies were collected (Supplementary Fig. 1).

### Evaluation of toxicity, dose modifications, and assessment of response
The adverse events were assessed by the NCI Common Terminology Criteria for Adverse Events (CTCAE) version 5. Dose-limiting toxicities (DLTs) were pre-specified study treatment-related adverse events: grade ≥3 immune-related colitis, grade 4 immune-related adverse event, grade ≥ 3 non-immune related adverse event with a pre-specified exception, delay in cisplatin greater than 2 weeks, or grade 5. The DLT period started with protocol therapy until 30 days after CRT (dose-limiting toxicities are further outlined in the protocol section 5.7). Any immune-related event that required immunosuppressive treatment or systemic steroids for ≥2 weeks was considered a DLT. Cisplatin was held for ANC < 1500/mcl and platelet count <100,000/mcl for a maximum of 3 weeks.

### T-Cell Repertoire immunosequencing
DNA was extracted from tumor biopsies and blood using Qiagen kits. Extracted genomic DNA was amplified in a bias-controlled multiplex PCR, followed by high-throughput sequencing using Adaptive Immunosequencing (Adaptive Biotechnologies, Seattle, WA) and the data analyses were performed according to the methods described previously[19–21]. Repertoire metrics include the total number of unique clones detected in tumor tissue and blood, quantification of clonality and diversity metrics, and quantification of the relative fraction of tumor-associated clones in peripheral blood. Shannon entropy was calculated from all productive TCR sequences and was normalized by dividing Shannon entropy by the logarithm of the number of all unique productive TCRs. Simpson clonality was calculated according to the formula $\sqrt{\Sigma(p_i)^2}$, where $p_i$ is the proportional abundance of clone i. Diversity (richness) in each sample was measured by counting the number of unique rearrangements after computationally downsampling to a common number of T cells. Tumor-associated TCR clones in peripheral blood were defined as follows: TCR clones detected in any tumor (pre- or post-treatment) and also detected in peripheral blood at any frequency. Peripheral clonal expansion was measured by calculating the differential abundance of clone frequencies between time points as described previously[22]. The clonal expansion was defined as clones that expanded or newly emerged from baseline to a later time point. Each clone was tested for significant expansion using a binomial test after correcting for the control false discovery rate (FDR) at 0.01[23]. The percentage of tumor-associated clones expanded in peripheral blood was calculated by dividing the number of different expanded tumor-associated clones by the total number of expanded clones.

### Pathology review
Pathologic confirmation of diagnosis was performed by each individual institution. Pathology scoring sheets as well as corresponding pathology reports were submitted. A study-dedicated pathologist at MSKCC performed confirmatory tissue evaluation.

## PD-L1 Immunohistochemistry

PD-L1 immunohistochemistry was done on pre-treatment tissue at CellCarta using the clinical SP263 assay with tumor cell and immune cell PD-L1 percentages reported.

## Statistical considerations

The primary objective was to determine whether there was a difference in TCR clonal expansion in peripheral blood at day 21 from baseline between Arm A and Arm B. The rationale for choosing day 21 as the timepoint for testing is based on prior translational studies of immune checkpoint blockade in other cancers, demonstrating that maximal T cell re-invigoration in peripheral blood happens within the first 3–6 weeks of therapy[24,25]. This justified the use of the day 21 time point, as it corresponded to 6 weeks of therapy in Arm A and 3 weeks of therapy in Arm B. The choice of day 21 also enabled the comparison of the evolution of TCR repertoires in response to a single dose of atezolizumab alone (day 0 in Arm A) vs. a single dose of atezolizumab with chemoradiation (day 21 in Arm B) as an exploratory parameter. TCR clonal parameters include the absolute number of total expanded TCR clones, absolute numbers of expanded tumor-associated TCR clones, and fraction of expanded tumor-associated clones out of total clones. Due to a lack of prior data, an optimal parameter was not known prior to the study's start. Since absolute clone numbers could be influenced by differences in baseline repertoires between the patients, the fraction of tumor-associated TCR clones that expand at day 21 from baseline was selected as the primary endpoint. The null hypothesis assumed no difference for the mean of TCR clonal expansion at day 21 from baseline between the two arms, and the alternative was the complement of the null with a clinically interesting effect size of 0.95, where the effect size is defined as the mean difference divided by the standard deviation. A sample size of 40 patients (20 in each arm) provided this study with a 90% power to detect a mean difference for a TCR clonal expression at day 21 between the two arms with an effect size of 0.95 at 10% significance level by a two-sided $t$-test under equal variance and normality assumptions. Due to attrition in patients or biospecimens, the final sample size of evaluable 17 patients in Arm A and 13 in Arm B gave the study 81% power to detect the effect size of 0.95 at the significance level of 0.1 using a two-sided $t$-test. For the detection of tumor-associated TCR clones, the total number of evaluable patients was 25 (14 in arm A and 11 in arm B), which gave the study 74% power to detect the effect size of 0.95 at the significance level of 0.1 using a two-sided $t$ test. A sensitivity analysis for the primary objective was conducted by a Wilcoxon rank sum test. All eligible patients who received atezolizumab on day -21 and day 0 for Arm A and day 0 for Arm B and had day 21 TCR measurements were included in the analysis for the primary objective by $t$ test.

Secondary endpoints in the study included the rate of DLTs, safety, 2-year DFS, and additional TCR repertoire parameters, including total clonal expansion, absolute counts of tumor-associated clonal expansion, clonality, and diversity. Evaluation of overall toxicity consisted of all patients who received any amount of protocol therapy, where DLT-evaluable patients were any eligible patients who received at least one dose of atezolizumab and had a DLT or completed protocol therapy. DFS was defined as the duration of time from study entry to a documented disease recurrence/progression or death, whichever occurred first. DFS was censored in patients who were alive without disease recurrence or progression. The 2-year DFS was a binary endpoint estimated using a binomial approach. If a patient survived disease-free for at least 2 years, this patient was considered to have 2-year DFS. Wilcoxon signed rank tests and Wilcoxon rank sum tests were performed for statistical comparisons within an arm and between the 2 arms[26], correspondingly. In general, for investigation of an association between two variables, chi-square tests or Fisher exact tests were employed for discrete-type variables, and Spearman's rank correlation coefficient tests were performed for continuous variables or a discrete-type variable with ordinal features[27,28]. Kaplan-Meier (KM) method was used to estimate the distributions for DFS and overall survival[29].

All tests were two-sided at 10% significance level, and no adjustments for multiple testing were planned for secondary and exploratory objectives due to their exploratory nature. SAS/STAT software (Version 9.4) was used in the analyses.

## Reporting summary

Further information on research design is available in the Nature Portfolio Reporting Summary linked to this article.

## Data availability

Individual participant data cannot be made publicly available due to patient privacy concerns. De-identified patient-level data will be made available to researchers with an approved NRG Oncology Data Use Agreement upon request to the NRG using the following email address: APC@NRGOncology.org. The NRG Data Use Agreement is in effect for up to 3 years. An extension can be pursued, or the data in all forms must be destroyed. Requestors may seek data with or without employment under, or affiliation with, an Entity (i.e., organization, institution, or university). If a requestor wishes to request data underemployment/affiliation with an Entity, all resources used within a research plan must belong to the Entity, and work performed must fall strictly within the scope of the requestor's employment/affiliation with the Entity. After a request is submitted and the Data Use Agreement is signed, the request is reviewed by NCI for adherence to legal and administrative requirements with a total timeframe of 2-3 months. The complete TCR sequencing data generated in this study are available for general research use and have been deposited in dbGAP with the following link: https://www.ncbi.nlm.nih.gov/projects/gap/cgi-bin/study.cgi?study_id=phs003833.v1.p1. These data are available under controlled access to be consistent with the informed consent of the original study participants. Access can be requested through dbGaP. The study protocol is available in the Supplementary Information file. The remaining data are available within the Article, Supplementary Information, or Source Data file. Source data are provided in this paper.

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

## Acknowledgements

This study was supported by National Cancer Institute grants to NRG Oncology U10CA180822 (NRG Oncology Statistics and Data Manage-ment Center), U10CA180868 (NRG Oncology Operations), and U24CA196067 (NRTG Oncology Biospecimen Bank). NRG-GY017 was sponsored by the National Cancer Institute (NCI) and was conducted by NRG Oncology under a Cooperative Research and Development Agree-ment (CRADA) between the NCI and Genentech. Biopsy funding was provided by Genentech to The GOG-Foundation, Inc. D.Z. is funded in part by the National Institutes of Health/National Cancer Institute Cancer Center Support Grant (P30-CA008748). D.Z. and J.M. received funding from the National Cancer Institute grant (R01CA276087). J.M. received funding from the National Cancer Institute grant (R50CA282102-01). The following NRG Oncology member institutions participated in the primary treatment studies: Georgia Cares Minority Underserved NCORP, UC San Diego Moores Cancer Center, University of Oklahoma Health Sciences Center LAPS, Women and Infants Hospital, Ohio State University Com-prehensive Cancer Center LAPS, UPMC Hillman Cancer Center LAPS, and Froedtert and the Medical College of Wisconsin LAPS. The authors thank Dr. Charles Kunos, MD, PhD, for expertize and guidance.

## Author contributions

J.M. and D.Z. contributed equally to this protocol and are listed as co-first authors. Jyoti Mayadev, Dmitriy Zamarin- design, accrual, safety monitoring, conduct of the trial, data collection, analysis, manuscript writing, manuscript review, manuscript editing. Wei Deng – design, conduct of the trial, data collection, analysis, manuscript writing, manuscript review, manuscript editing. Heather A Lankes - conduct of the trial, data collection, biobank quality, manuscript writing, manu-script review, manuscript editing. Giulio Pesci - data collection, analysis, manuscript review, manuscript editing. Hayeon Kim- physics oversight, manuscript review, manuscript editing. Junzo P. Chino- radiation over-sight, data collection, analysis, manuscript writing, manuscript review, manuscript editing. Barbara Banbury - analysis, manuscript writing, manuscript review, manuscript editing. Ned Sherry- analysis, manu-script writing, manuscript review, manuscript editing. Elad Sharon Sharad A. Ghamande, Catherine Ferguson, Loren Mell, Laura Holman, Cara Mathews, David O'Malley, Alexander Olawaiye, Elizabeth Hopp, Charles Leath III, Larry Copeland, Robert Mannel, Roisin O'Cearbhaill, Carol Aghajanian: Accrual, data collection, analysis, manuscript writing, manuscript review, manuscript editing. Russell J. Schilder – design input, conduct of the trial, data collection, analysis, manuscript writing, manuscript review, manuscript editing.

## Competing interests

Jyoti Mayadev reports personal fees from AstraZeneca, Merck, Prim-mune, Varian Medical Systems, The GOG Foundation, Inc; grant support from NCI, NRG Oncology, Moores Cancer Center UCSD, The GOG Foundation, Inc. Dmitriy Zamarin reports institutional grants from Merck, Genentech, AstraZeneca, Plexxikon, and Synthekine, and personal fees from AstraZeneca, Xencor, Memgen, Takeda, Synthekine, Immunos, Tessa Therapeutics, Miltenyi, and Calidi Biotherapeutics. DZ owns a patent on the use of oncolytic Newcastle Disease Virus for cancer therapy. Junzo Chino received personal consulting fees from Stryker. He also received a stipend for chapter writing (personal) from the GOG Foundation. He participated on a Data Safety Monitoring Board for KM Pharmaceutical Consulting LLC. He also served as an unpaid Board Member for the American Brachytherapy Society. Barbara Banbury is employed by and holds stock in Adaptive Biotechnologies. Ned Sherry is employed by and holds stock in Adaptive Biotechnologies. Sharad Ghamande received clinical trial payments to the institution from Merck, GSK/Tejaro, Jovance, Clovis, Oncology, Takedo, and Eisai. He received payment or honoraria for lectures, presentations, speaker bureaus, manuscript writing, or educational events from GSK and Eisai. Loren

Mell's institution received grants or contracts from Merck and AstraZeneca. He also received consulting fees from Cel-Sci as well as payment for expert testimony from Sanofi. He participated on a Data Safety Monitoring Board for Pfizer. Cara Mathews' institution received funding from the National Cancer Institute. She also received grants or contracts from Syros, Deciphera, Astellas Pharma, Seagen, Genmab, EMD Serono, Merck, Regeneron, Moderna, AstraZeneca, AvengeBio, Zentalis, GlaxoSmithKline, and Genentech outside the submitted work. David O'Malley received support from the NRG/NCI/GOG Foundation. He also received grants or contractd from AbbVie, Advaxis, Agenus, Inc., Alkermes, Aravive, Inc., Arcus Biosciences, Inc., AstraZeneca, BeiGene USA, Inc., Boston Biomedical, Bristol Myers Squibb, Clovis Oncology, Deciphera Pharma, Eisai, EMD Serono, Inc., Exelixis, Genentech Inc., Genmab, GlaxoSmithKline, GOG Foundation, Hoffmann-LaRoche Inc., ImmunoGen, Inc., Incyte Corporation, IOVANCE, Biotherapeutics, Karyopharm, Leap Therapeutics, Inc., Ludwig Institute for Ca, Merck & Co., Merck Sharp and Dohme Corp., Mersana Therapeutics, Inc., NCI, Novartis, Novocure, NRG Oncology, OncoC4, Inc., OncoQuest Inc., Pfizer Inc., Precision Therapeutics, Inc., Prelude Therapeutics, Regeneron Pharmaceuticals, Inc., RTOG, Rubius Therapeutics, Seattle Genetics (SeaGen), Sutro BioPharma, SWOG, TESARO, Verastem, Inc. Dr. O'Malley reports personal fees from Consulting and/or advisory board member from AbbVie, AdaptImmune, Agenus,Inc, Arquer Diagnostics, Arcus Biosciences, Inc., AstraZeneca, Atossa Therapeutics, Boston Biomedical, Cardiff Oncology, Celcuity, Clovis Oncology, Corcept Therapeutics, Duality Bio, Eisai, Elevar, Exelixis, Genentech Inc, Genelux, GlaxoSmithKline, GOG Foundation, Hoffmann-La Roche Inc, ImmunoGen, Inc, Imvax, InterVenn, INXMED, IOVANCE Biotherapeutics, Janssen, Jazz Pharmaceuticals, Laekna, Leap Therapeutics, Inc., Luzsana Biotechology, Merck & Co, Merck Sharp & Dohme Corp, Mersana Therapeutics,Inc, Myriad, Novartis, NovoCure, OncoC4, Inc., Onconova, Regeneron Pharmaceuticals, Inc, RepImmune, R Pharm, Roche Diagnostics, Seattle Genetics (SeaGen), Sorrento, Sutro Biopharma, Tarveda Therapeutics, Toray, Trillium, Umoja, Verastem, Inc, VBL Therapeutics, Vincerx Pharma, Xencor, and Zentalis. Alexander Olawaiye received an Honorarium for Advisory Board Meetings from AstraZeneca, GSK, and Merck. He also received a grant payment made to his institution from AstraZeneca. Elizabeth Hopp participated in the Immunogen Advisory Board on 6/1/23. Charles Leath contracted research with AstraZeneca and Merck. He also received a consulting/Scientific Advisory Board with Merck and Seattle Genetics. Larry Copeland received payment to himself, with institutional approval from the GOG Foundation President. He also received support for travel to GOG Semiannual meetings based on receipts. Robert Mannel received payment from his institution for NCI/NRG trial-based capitation. Roisin O'Cearbhaill received support from NCI/NIH (grant P30 CA008748). She also received grants or contracts paid to her institution from: Bayer/Celgene/Juno, Tesaro/GSK, Merck, Ludwig Cancer Institute, Abbvie/StemCentrx, Regeneron, TCR2 Therapeutics, Atara Biotherapeutics, Marker Therapeutics, Syndax Pharmaceuticals, Genmab/Seagen Therapeutics, Genentech, Alkermes, Kite Pharma, Acrivon and Gynecologic Oncology Foundation, Lyell Immunopharma, Bayer/Celgene/Juno, Tesaro/GSK, Merck, Ludwig Cancer Institute, Abbvie/StemCentrx, Regeneron, TCR2 Therapeutics, Atara Biotherapeutics, Marker Therapeutics, Syndax Pharmaceuticals, Genmab/Seagen Therapeutics, Genentech, Alkermes, Kite Pharma, Acrivon and Gynecologic Oncology Foundation, Lyell Immunopharma. She also received payment or honoraria for lectures, presentations, speaker bureaus, manuscript writing, or educational events from GSK, Ciro/Onclive/PER/MJH/Aptitude Health, SITC, and Gynecologic Oncology

Canada. She received support for attending meetings and/or travel from Hitech Health, Gathering Around Cancer, Ireland, GOG Foundation, and SGO. She participated on a Data Safety Monitoring Board or Advisory Board from AstraZeneca (DUO-0), GSK (Moonstone, Prma) and Acrivon for unpaid steering committee, Carina Biotech, Link Therapeutics, Tesaro/GSK, Regeneron Advisory, Seattle Genetics/SeaGen, Immunogen board, Bayer, R-Pharm, Fresenius Kabi, Miltenyi, 2Seventybio and Bayer for advisory boards. She had a leadership or fiduciary role as Vice-Chair for CPC, SGO Chair, CT Committee, and NRG Oncology. Carol Aghajanian received clinical trial funding to her institution (MSK) as follows: Abbvie – MSK, PI, GOG 3005, AstraZeneca – MSK PI, SOLO1/GOG 3004, National Coordinating Investigator & MSK PI, DO81RC00001; ENGOT – ov 46; AGO-OVAR 23; GOG–3025; Clovis – MSK PI, ARIEL 2 & 3; Genentech/Roche – MSK PI, GOG 3015 (IMagyn050). She also received consulting fees from Roche/Genentech – Advisory Board 8/21/20; Eisai/Merck – Advisory Board 9/12/20; AstraZeneca/Merck – Advisory Boards 9/30/20 & 10/14/20 and Repare Therapeutics – Advisory Board 10/15/20. She participated on an Advisory Board 6/30/21 for Blueprint Medicine. She served on GOG Foundation, Board of Directors (unpaid, occasional travel cost reimbursement to attending meetings); NRG Oncology Board of Directors (unpaid). Russell Schilder received consulting fees from Incyte, and Honoraria from Pfizer and participated on a Data Safety Monitoring Board or Advisory Board for Celsion. All other co-authors have no Competing Interests to declare.

## Additional information

[1]University of California San Diego, San Diego, USA. [2]Icahn School of Medicine at Mount Sinai, New York City, USA. [3]NRG Oncology Statistics & Data Center, Buffalo, USA. [4]NRG OncologyOperations Center-Philadelphia East, Philadelphia, USA. [5]Memorial Sloan-Kettering Cancer Center, New York City, USA. [6]UPMC Hillman Cancer Center, Pittsburgh, USA. [7]Duke University, Durham, USA. [8]Adaptive Biotechnologies Corp, Seattle, USA. [9]Dana Faber Cancer Institute,

Boston, USA. [10]Augusta University Medical College of Georgia, Augusta, USA. [11]University of Oklahoma Health Sciences Center, Oklahoma City, USA. [12]Women & Infants Hospital, Providence, USA. [13]The Ohio State University Wexner Medical Center Columbus, Columbus, USA. [14]University of Pittsburgh Medical Center, Pittsburgh, USA. [15]Medical College of Wisconsin, Milwaukee, USA. [16]University of Alabama, Birmingham, USA. [17]Jefferson University Sidney Kimmel Medical College, Philadelphia, USA. [18]These authors contributed equally: Jyoti Mayadev, Dmitriy Zamarin. ✉e-mail: jmayadev@health.ucsd.edu

