## [Peer Review File · Nature Communications]

Neoadjuvant or concurrent atezolizumab with
chemoradiation for locally advanced cervical cancer: a
randomized phase I trialREVIEWER COMMENTS

Reviewer #1 (Remarks to the Author): with expertise in computational immunology, TCR repertoire

This paper reports a small clinical trial exploring the time-dependence of combining chemoradiation and anti-PD1 treatment on advanced cervical cancer. The principal output were global TCR repertoire metrics. The major findings were that a predose of anti-PD1 before radiation gave rise to an increased proportion and expansion of tumour TCRs in blood after cycle 1, and sustained. There was also a small improvement in DFS. Overall, this study is clearly described (although I found the description of the two arms quite hard to follow) and the results are convincing. I am not convinced the study adds much to our mechanistic understanding of the interaction between immunotherapy and radiation, and I feel the paper is more suited to a more clinical journal than to the broad readership of Nature Comms.

Reviewer #2 (Remarks to the Author): with expertise in biostatistics, clinical trial study design

The study explored the sequencing of atezolizumab (an immune checkpoint inhibitor) with chemoradiation therapy (CRT) in patients with high-risk node-positive locally advanced cervical cancer (LACC). Key findings include a higher proportion of tumor-associated T-cell receptor (TCR) clones expansion in peripheral blood with neoadjuvant atezolizumab prior to CRT compared to concurrent administration. There was a trend towards improved 2-year disease-free survival (DFS) in the neoadjuvant atezolizumab arm (76% vs. 56% in the concurrent arm), although not statistically significant.

The study's sample size was smaller than targeted, potentially limiting the statistical power to detect significant differences in clinical outcomes. Meanwhile, the translational sample size attrition, although pre-specified, may introduce bias and affect the generalizability of the findings.

The analysis method is sound, with well-defined treatment protocols and endpoints.

However, the smaller than targeted sample size and potential biases from sample attrition should be acknowledged.

Reviewer #3 (Remarks to the Author): with expertise in gynecological cancers, radiotherapy/immunotherapy

Mayadev & Zamarin et al. present the results of a randomized phase I trial involving the sequential administration of atezolizumab (an anti-PDL-1 inhibitor) and chemoradiotherapy (CRT) in patients with high-risk node-positive locally advanced cervical cancer (LACC). The primary objective was to evaluate the expansion of tumor-associated TCR clones in peripheral blood as a surrogate measure of immune response, with the secondary objectives of safety, tolerability, and 2-year disease free survival (DFS).

This study is both timely and innovative, given the recent findings from the randomized trial KeyNote A-18, which demonstrated improved progression-free survival (HR 0.7; 95% CI 0.55-0.8; p=0.002) with pembrolizumab (an anti-PD1 inhibitor administered at 200 mg Q3W) concurrently with CRT, followed by adjuvant pembrolizumab (400 mg Q6W for 15 cycles) after CRT (Lorusso Lancet 2024). The observed overall survival rate at 24 months was 87% in the pembrolizumab–CRT group and 81% in the placebo–CRT group, indicating a significant clinical benefit. (Lorusso Lancet 2024)

My general suggestion is to enhance this publication by incorporating additional translational data derived from tumor biopsies. This could provide valuable insights into the mechanistic underpinnings of the observed treatment responses and help elucidate potential biomarkers predictive of patient outcomes. By integrating such translational data, we can further deepen our understanding of the molecular mechanisms driving therapeutic efficacy and potentially identify novel targets for intervention or refinement of treatment strategies.

Introduction:

Line 90: a slight typo is present “(CRT)have”

Materials and Methods

1. Given that this was a phase I study with safety as a secondary endpoint, it would be beneficial for the authors to provide clarification on several key aspects of patient enrollment and trial conduct. Specifically, elucidating the process of patient recruitment, including any potential waiting periods between patients, would offer valuable insight into the study's methodology. Additionally, detailing how dose-limiting toxicities (DLTs) were defined and the criteria for halting the trial in the event of safety concerns would enhance understanding of the safety monitoring procedures implemented. Regarding the primary objective, while the selection of T-cell receptor (TCR) clonal expansion at day 21 as a surrogate endpoint of efficacy appears appropriate, it would be advantageous for the authors to provide further context regarding this choice. As a non-statistician, I find this approach elegant; however, additional information on treatment and trial decisions, particularly within the phase I context, would provide a more comprehensive understanding of the study's rationale and execution. Such insights could further underscore the significance and relevance of the findings presented.

2. Line 147 Small typo : parametersinclude

3. Clarification regarding the definition and characterization of tumor-associated T-cell receptor (TCR) populations would be valuable to enhance the comprehensibility of the Methods section. Specifically, detailing the metrics and methodologies employed, such as Gini coefficient or entropy analysis, to identify and quantify these TCRs would provide greater clarity and transparency regarding the analytical approach utilized. Moreover, incorporating this information into the Results section could enrich the interpretation of the findings, offering insights into the diversity and distribution of tumor-infiltrating TCR populations. By elucidating the analytical techniques utilized and their outcomes, the authors can provide a more robust description of the immune response dynamics within the tumor microenvironment, thereby augmenting the overall scientific rigor and impact of the study.

Results:

1. Table 1: There are notable imbalances in patient selection between arms A and B. For instance, Arm A exhibits a higher proportion of adenocarcinoma tumors (21%) compared to arm B (5.9%). Additionally, the presence of parametrial involvement (PAO+) was observed in 47% of patients in arm A and 29% in arm B. Given that these factors are widely recognized as adverse prognostic indicators typically utilized for stratification, I kindly request clarification from the authors regarding the specific stratification factors employed for randomization.

2. The radiation doses, brachytherapy doses, and median duration of radiation per arm are not presented in any of the tables

3. Line 203: The median follow-up duration of the trial (25.8 months) is deemed appropriate, considering that recurrences typically manifest early, typically within the initial 2-3 years post-treatment. It's noteworthy that there were no significant differences observed in median disease-free survival (DFS) and overall survival (OS) between the arms. However, it's important to add a few words clarifying that the trial was not powered to detect significant differences in its secondary endpoints.

I recommend transferring the OS data (Figure) to the Supplementary Material and retaining the DFS data in the main figure. Additionally, it would be beneficial to include a schematic representation of the trial's therapeutic regimen in the figure for enhanced clarity and understanding.

4. Figure 3 F. Could the authors separate the patients in arm A responders vs Arm A non-responders and do the same analysis? Same for arm B responders vs non-responders.

5. As a considerable portion of the patients exhibited PDL1 staining, it would be beneficial to include pathology images in the manuscript. Similarly, I suggest that the authors explore the potential association between the presence of PDL1 and TCR specificity as a surrogate marker for precursor exhausted T cells. This analysis could provide valuable insights into the immune landscape of the tumor microenvironment and further elucidate the mechanisms underlying treatment response.

6. As tumor-specific T-cell receptors (TCRs) serve as a reflection of the presence or absence of T cells, with the latter being a more readily accessible pathological biomarker (albeit not yet validated in cervical cancer), I kindly request the authors to furnish data on the presence of CD8+PD1+ cells within the tumor microenvironment (TME) of responders and non-responders in both arms A and B. This analysis would offer valuable insights into the immune landscape of the TME and may provide clues regarding treatment response mechanisms. Adding co-staining with Ki67 or Granzyme B for instance would be of high value.

7. Did radiation therapy contract the TCR repertoire? Typically, radiation is associated with a decrease in white blood cells and more specifically lymphocytes in peripheral blood, particularly notable if parametrial involvement (PAO) irradiation was conducted (several radiation papers show this). Therefore, I suggest that the authors conduct a correlation analysis between the timing of white blood cell reduction during radiation and the disappearance of TCR clones. In essence, this analysis would provide an idea whether the observed decrease in TCR clones is attributable to the radiation-chemo-induced tumor shrinkage or direct damage to tumor-specific T cells within the tumor microenvironment.

Discussion

The discussion is well-articulated. I would recommend addressing the limitations arising from the imbalanced distribution of certain factors between the arms, notably:

- Adenocarcinoma: 21% in arm A vs. 5.9% in arm B
- Squamous cell carcinoma (SCC): 3.7% in arm A vs. 82% in arm B
- Parametrial involvement (PAO+): 47% in arm A vs. 29% in arm B
- PDL1 expression: 42% in arm A vs. 53% in arm B

Despite these inherent imbalances expected in a small translational protocol, it's noteworthy that arm A, which comprised patients with potentially worse prognostic factors and exhibited the lowest rates of radiotherapy completion (particularly brachytherapy), demonstrated superior outcomes. Specifically, a higher proportion of pathological responses were observed at 21 days (69% arm A vs. 40% arm B), and 68% of patients in arm

A were disease-free at 2 years, compared to 52% in arm B, which although not statistically significant it was numerically higher.

This places the findings of this small translational study in perspective with the results of KEYNOTE A18, which evaluated concurrent and adjuvant pembrolizumab. Notably, the largest benefit in KEYNOTE A18 was observed in stage III-IVa disease, with the majority of patients (90%) being PDL1 positive and limited benefit observed for adenocarcinomas. The NRG-GY017 study could serve as a foundation for exploring the potential of neoadjuvant immunotherapy in PDL1-negative, locally advanced adenocarcinomas, for which treatment options are currently limited.

Reviewer #4 (Remarks to the Author): with expertise in cervical cancer, (immuno)therapy

Jyoti Mayadev and colleagues conducted a phase I trial evaluating neoadjuvant or concurrent Atezolizumab with chemoradiation for locally advanced cervical cancer with 40 patients enrolled. The discussion on optimal sequencing of CRT and ICB is an interesting topic. However, the issues below should be properly addressed

Major concerns

1. Due to the fact that cervical cancer is a chemo-sensitive tumor, there is marginal space for increasing response rates. The expansion of tumor-associated TCR (as a surrogate for anti-tumor immune response) in this study did not seem to translate into prognosis advantage. Furthermore, the TCR expansion advantage did not persist upon day 21. Please comment.
2. The manuscript lacks a clear definition of "tumor-associated clone." Given the importance of this concept to the study's objectives, it is essential to provide a precise definition to ensure basic understanding. Please define the term.
3. According to the patient demographics and characteristics, patients in Arm A demonstrated significantly higher pre-treatment PD-L1 (SP263) tumor cell score compared with that of Arm B. Could it be important confounding factor in results interpretation? Please elaborate on this.
4. Line 117, explain IMRT

5. Line 148 Typo, "randomized"

1. Line 184 Typo, "parametersinclude"

REVIEWER COMMENTS

Reviewer #1 (Remarks to the Author): with expertise in computational immunology, TCR repertoire

1. This paper reports a small clinical trial exploring the time-dependence of combining chemoradiation and anti-PD1 treatment on advanced cervical cancer. The principal output were global TCR repertoire metrics. The major findings were that a predose of anti-PD1 before radiation gave rise to an increased proportion and expansion of tumour TCRs in blood after cycle 1, and sustained. There was also a small improvement in DFS. Overall, this study is clearly described (although I found the description of the two arms quite hard to follow) and the results are convincing. I am not convinced the study adds much to our mechanistic understanding of the interaction between immunotherapy and radiation, and I feel the paper is more suited to a more clinical journal than to the broad readership of Nature Comms.

1. Response:

We appreciate the comments and time to review the paper. We have edited the manuscript to hopefully make the arms easier to follow. To our knowledge, this study presents the first prospective data in the literature to examine the role of induction immunotherapy prior to chemoradiation with immunotherapy with investigations into the TCR repertoire evolution. We think that these data are quite timely in the reporting as the KEYNOTE A18 manuscript published in the spring of 2024 demonstrated a marginal, albeit statistically significant DFS benefit to concurrent pembrolizumab with chemoradiation in high risk cervical cancer leading to the recent FDA approval in the United States. Aside from the KEYNOTE A18 trial, other trials of concurrent immunotherapy and chemoradiation in cervical and head and neck cancers have been negative. The question of optimal timing of immunotherapy with chemoradiation is thus a hot topic and, given the signals of clinical and immune activity in our study, we do feel that our findings would be of interest to the broad readership of Nature Communications and will generate rationale for trial designs using neoadjuvant approaches across a number of cancers. Of note, when this study was initially designed in 2016-2017 with the NCI, the approach for cervical cancer was quite novel to prime the immune system prior to full dose chemoradiation. At ASCO 2023 and 2024 there were several global abstracts in cervical cancer examining pre-RT chemotherapy and immunotherapy so there continues to be interest and clinical trials examining this approach. We now present our mature study findings in Nature Communications, and appreciate the review and expertise.

Reviewer #2 (Remarks to the Author): with expertise in biostatistics, clinical trial study design

The study explored the sequencing of atezolizumab (an immune checkpoint inhibitor) with chemoradiation therapy (CRT) in patients with high-risk node-positive locally advanced

cervical cancer (LACC). Key findings include a higher proportion of tumor-associated T-cell receptor (TCR) clones expansion in peripheral blood with neoadjuvant atezolizumab prior to CRT compared to concurrent administration. There was a trend towards improved 2-year disease-free survival (DFS) in the neoadjuvant atezolizumab arm (76% vs. 56% in the concurrent arm), although not statistically significant.

2. The study's sample size was smaller than targeted, potentially limiting the statistical power to detect significant differences in clinical outcomes. Meanwhile, the translational sample size attrition, although pre-specified, may introduce bias and affect the generalizability of the findings. The analysis method is sound, with well-defined treatment protocols and endpoints. However, the smaller than targeted sample size and potential biases from sample attrition should be acknowledged.

2. Response:

We appreciate the reviewer's suggestion and fully agree with the concern. Despite the fact that our power calculations have accounted for the translational sample size attrition, we have highlighted that the small sample size, although pre-specified, may introduce bias and affect the generalizability of the findings. We have highlighted the study limitation due to the size throughout the manuscript and added the statement above to the limitation section of the discussion.

Reviewer #3 (Remarks to the Author): with expertise in gynecological cancers, radiotherapy/immunotherapy

Mayadev & Zamarin et al. present the results of a randomized phase I trial involving the sequential administration of atezolizumab (an anti-PDL-1 inhibitor) and chemoradiotherapy (CRT) in patients with high-risk node-positive locally advanced cervical cancer (LACC). The primary objective was to evaluate the expansion of tumor-associated TCR clones in peripheral blood as a surrogate measure of immune response, with the secondary objectives of safety, tolerability, and 2-year disease free survival (DFS).

This study is both timely and innovative, given the recent findings from the randomized trial KeyNote A-18, which demonstrated improved progression-free survival (HR 0.7; 95% CI 0.55-0.8; p=0.002) with pembrolizumab (an anti-PD1 inhibitor administered at 200 mg Q3W) concurrently with CRT, followed by adjuvant pembrolizumab (400 mg Q6W for 15 cycles) after CRT (Lorusso Lancet 2024). The observed overall survival rate at 24 months was 87% in the pembrolizumab-CRT group and 81% in the placebo-CRT group, indicating a significant clinical benefit. (Lorusso Lancet 2024)

3. My general suggestion is to enhance this publication by incorporating additional translational data derived from tumor biopsies.

3. Response:

We thank the reviewer for the kind review and enthusiasm and the suggestion to integrate additional translational data. We fully agree that the study has generated a rich resource of biospecimens and the current translational plan for the trial includes additional analyses of the tumor microenvironment pre- and post-treatment as well as evolution of circulating tumor DNA. Unfortunately, the rules stipulated by NCI-CTEP that sponsored this study require publication of primary trial endpoints (clinical and translational) before additional exploratory translational studies could be initiated: i.e. per NCI rules, they will not even release the additional biospecimens to us until we publish the current data.

Nevertheless, we feel that the clinical and translational data that have already been generated and included in this report provide novel and valuable insights into the biology of sequencing of immunotherapy and chemoradiation that could guide further trial development in the neoadjuvant space in cervical and other cancers. We do plan to perform and report on further translational research from this trial and have secured funding to do the work.

4. Introduction:

Line 90: a slight typo is present “(CRT)have”

4. *Response:*

Thank you. We have corrected this typo.

Materials and Methods

5. Given that this was a phase I study with safety as a secondary endpoint, it would be beneficial for the authors to provide clarification on several key aspects of patient enrollment and trial conduct. Specifically, elucidating the process of patient recruitment, including any potential waiting periods between patients, would offer valuable insight into the study's methodology.

5. *Response:*

Thank you. The patients were enrolled in a consecutive manner with a safety monitoring rule dictating a study hold in the event of DLTs observed in more than 30% DLT-evaluable patients in either one arm and although this was a phase I study, we did not have any safety pauses to enrollment on this trial. The patients were recruited from the NRG Oncology phase I GYN roster which is a specific site roster with expertise in phase I trials in gynecology oncology cancers. We have added this to the methods section.

6. Additionally, detailing how dose-limiting toxicities (DLTs) were defined and the criteria for halting the trial in the event of safety concerns would enhance understanding of the safety monitoring procedures implemented.

6. Response:

Thank you. We have added the following statement to the methods section so that the reader can see the full details of the DLT definitions in the study protocol:

“Evaluation of toxicity, Dose Modifications, and Assessment of Response. The adverse events were assessed by the NCI Common Terminology Criteria for Adverse Events (CTCAE) version 5. Dose limiting toxicities (DLTs) were pre-specified study treatment related adverse events: grade ≥ 3 immune-related colitis, grade 4 immune-related adverse event, grade ≥ 3 non-immune related adverse event with a pre-specified exception, delay in cisplatin greater than 2 weeks, or grade 5. The DLT period started with protocol therapy until 30 days after CRT (dose limiting toxicities are further outlined in the protocol section 5.7). Any immune-related event that required immunosuppressive treatment or systemic steroids for ≥ 2 weeks was considered a DLT. Cisplatin was held for ANC $< 1,500/\text{mcl}$ and platelet count $< 100,000/\text{mcl}$ for a maximum of 3 weeks”.

7. Regarding the primary objective, while the selection of T-cell receptor (TCR) clonal expansion at day 21 as a surrogate endpoint of efficacy appears appropriate, it would be advantageous for the authors to provide further context regarding this choice. As a non-statistician, I find this approach elegant; however, additional information on treatment and trial decisions, particularly within the phase I context, would provide a more comprehensive understanding of the study's rationale and execution. Such insights could further underscore the significance and relevance of the findings presented.

7. Response:

We thank the reviewer for the comment. In truth, dynamic biomarkers of immune response in peripheral blood have not been previously explored in the setting of chemoradiation. Our rationale for choosing day 21 as the timepoint for testing is based on prior translational studies of immune checkpoint blockade in other cancers, demonstrating that maximal T cell re-invigoration in peripheral blood happens within the first 3-6 weeks on therapy. This justified the use of day 21 time point, as it corresponded to 6 weeks of therapy in Arm A and 3 weeks in Arm B. The choice of day 21 also enabled us to compare the evolution of TCR repertoires in response to single dose of atezolizumab alone (day 0 in Arm A) vs. single dose of atezolizumab with chemoradiation (day 21 in Arm B). We have included this rationale in the manuscript in the Methods section.

8. Line 147 Small typo : parameters include

8. Response:

Corrected

9. Clarification regarding the definition and characterization of tumor-associated T-cell receptor (TCR) populations would be valuable to enhance the comprehensibility of the Methods section. Specifically, detailing the metrics and methodologies employed, such as Gini coefficient or entropy analysis, to identify and quantify these TCRs would provide

greater clarity and transparency regarding the analytical approach utilized. Moreover, incorporating this information into the Results section could enrich the interpretation of the findings, offering insights into the diversity and distribution of tumor-infiltrating TCR populations. By elucidating the analytical techniques utilized and their outcomes, the authors can provide a more robust description of the immune response dynamics within the tumor microenvironment, thereby augmenting the overall scientific rigor and impact of the study.

9. Response:

We thank the reviewer for these comments. While in favor of space conservation we have predominantly relied on referencing the analysis methodologies from prior studies, we agree that addition of details can aid the readers with understanding of the rationale and interpretation of the findings. We have modified the manuscript text throughout per the reviewer suggestions which hopefully has improved the clarity and transparency.

Results:

10. Table 1: There are notable imbalances in patient selection between arms A and B. For instance, Arm A exhibits a higher proportion of adenocarcinoma tumors (21%) compared to arm B (5.9%). Additionally, the presence of parametrial involvement (PAO+) was observed in 47% of patients in arm A and 29% in arm B. Given that these factors are widely recognized as adverse prognostic indicators typically utilized for stratification, I kindly request clarification from the authors regarding the specific stratification factors employed for randomization.

10. Response:

We appreciate the keen observation from the reviewer. In this phase I study we did not upfront stratify within the high-risk population that was enrolled on this study. As shown in Table 1, we agree that Arm A is enriched for patients with that exhibit potentially worse clinical and pathological factors. The squamous cell carcinoma was 73.7% in Arm A vs 82% in Arm B as indicated in table 1 patient characteristics and demographics. Nevertheless, even with the potentially inferior prognostic characteristics, patients in Arm A experienced a higher 2- year DFS.

To highlight the reviewer's point, we have edited the results and discussion sections to emphasize this imbalance.

11. The radiation doses, brachytherapy doses, and median duration of radiation per arm are not presented in any of the tables

11. Response:

Thank you. We have provided additional information in the results section, line 165:

“(34/36) completed EBRT per protocol of 45Gy with an external beam boost to the involved lymph nodes to 54-58Gy depending on location for brachytherapy contribution to the lymph nodes, and 94% (34/36) received brachytherapy with a prescription dose of 6-7Gy in 4-5 fractions (please see radiation section of the protocol in the appendix for further radiation prescription details).”

Due to the length of the radiation prescription, detailed description of the dose metrics and brachytherapy prescriptions we included the protocol which has this section in detail in the appendix.

12. Line 203: The median follow-up duration of the trial (25.8 months) is deemed appropriate, considering that recurrences typically manifest early, typically within the initial 2-3 years post-treatment. It's noteworthy that there were no significant differences observed in median disease-free survival (DFS) and overall survival (OS) between the arms. However, it's important to add a few words clarifying that the trial was not powered to detect significant differences in its secondary endpoints.

12. Response:

We completely agree and have added the following into the results section:

It should be noted that the trial was not powered to detect significant differences in its secondary endpoints between the study Arms.

13. I recommend transferring the OS data (Figure) to the Supplementary Material and retaining the DFS data in the main figure. Additionally, it would be beneficial to include a schematic representation of the trial's therapeutic regimen in the figure for enhanced clarity and understanding.

13. Response:

Thank you. Given your recommendations, we will take out Figure 2B as the OS and move this to supplementary figure 1 as the KM curve for OS. We have included a schematic representation of the therapeutic regimen in figure 1.

14. Figure 3 F. Could the authors separate the patients in arm A responders vs Arm A non-responders and do the same analysis? Same for arm B responders vs non-responders.

14. Response:

Thank you for this insightful comment. Following the reviewer's suggestion, we performed the analysis of the association between tumor-associated clonal expansion and 2-year DFS in each arm. We present these data as Supplementary Figure 5C. Unfortunately, no solid conclusions could be reached based on this analysis, likely due to small number of events in each arm.

15. As a considerable portion of the patients exhibited PDL1 staining, it would be beneficial to include pathology images in the manuscript. Similarly, I suggest that the authors explore the potential association between the presence of PDL1 and TCR specificity as a surrogate marker for precursor exhausted T cells. This analysis could provide valuable insights into the immune landscape of the tumor microenvironment and further elucidate the mechanisms underlying treatment response.

15. Response:

We thank the reviewer for this comment. We have now added the PD-L1 analyses as a function of time point (Supplementary Figure 3) demonstrating some increase in PD-L1 staining in response to therapy although this was not consistent across all patients. Unfortunately, there was a significant number of patients with missing PD-L1 data who also had complete TCR data, which precluded definitive association analyses between PD-L1 positivity and evolution of TCR repertoires. We indeed performed such analyses but did not identify any association between the baseline PD-L1 status and TCR parameters. We have mentioned this lack of association in the manuscript (lines 226-248). As the reviewer has pointed out, it would be important to examine T cell phenotypes (i.e. precursor exhausted T cells) in the tumor microenvironment and their association with PD-L1 expression. Unfortunately, as outlined above, we are currently limited in the types of exploratory analyses that we can carry out before the primary outcomes data are published. The analysis of these parameters is certainly planned!

16. As tumor-specific T-cell receptors (TCRs) serve as a reflection of the presence or absence of T cells, with the latter being a more readily accessible pathological biomarker (albeit not yet validated in cervical cancer), I kindly request the authors to furnish data on the presence of CD8+PD1+ cells within the tumor microenvironment (TME) of responders and non-responders in both arms A and B. This analysis would offer valuable insights into the immune landscape of the TME and may provide clues regarding treatment response mechanisms. Adding co-staining with Ki67 or Granzyme B for instance would be of high value.

16. Response:

We completely agree with the reviewer about the importance of these data, but once again would like to highlight that we are currently restricted in such analyses by NCI-imposed rules. We want to emphasize that we plan on further work and reporting of these important translational data.

17. Did radiation therapy contract the TCR repertoire? Typically, radiation is associated with a decrease in white blood cells and more specifically lymphocytes in peripheral blood, particularly notable if parametrial involvement (PAO) irradiation was conducted (several radiation papers show this). Therefore, I suggest that the authors conduct a correlation

analysis between the timing of white blood cell reduction during radiation and the disappearance of TCR clones. In essence, this analysis would provide an idea whether the observed decrease in TCR clones is attributable to the radiation-chemo-induced tumor shrinkage or direct damage to tumor-specific T cells within the tumor microenvironment.

17. Response:

We thank the reviewer for the excellent suggestion. We have performed the suggested analyses by analyzing the changes in WBC parameters. Indeed, contraction of the TCR repertoire during radiation strongly correlated with reduction in WBC numbers. Since the majority of the TCRs in peripheral blood are not tumor-associated, we believe that this contraction is likely related to the radiation-induced general lymphopenia rather than specific killing of tumor-specific T cells within the TME and tumor-draining lymph nodes. However, with this in mind, we also observe relative reduction of tumor-associated TCR clones in peripheral blood (out of total clones) in response to radiation, particularly in Arm A where these clones have previously expanded. This would suggest that radiation could indeed have negative impact on expansion of tumor-associated TCR clones, possibly through their direct killing in the tumor-draining lymph nodes. We have included this observation in the manuscript.

Discussion

18. The discussion is well-articulated. I would recommend addressing the limitations arising from the imbalanced distribution of certain factors between the arms, notably:

- Adenocarcinoma: 21% in arm A vs. 5.9% in arm B
- Squamous cell carcinoma (SCC): 73.7% in arm A vs. 82% in arm B
- Parametrial involvement (PAO+): 47% in arm A vs. 29% in arm B
- PDL1 expression: 42% in arm A vs. 53% in arm B

Despite these inherent imbalances expected in a small translational protocol, it's noteworthy that arm A, which comprised patients with potentially worse prognostic factors and exhibited the lowest rates of radiotherapy completion (particularly brachytherapy), demonstrated superior outcomes. Specifically, a higher proportion of pathological responses were observed at 21 days (69% arm A vs. 40% arm B), and 68% of patients in arm A were disease-free at 2 years, compared to 52% in arm B, which although not statistically significant it was numerically higher.

18. Response:

Thank you. We agree that Arm A has potentially worse clinical and pathological factors, and yet experiences a higher 2 yr. DFS. The squamous cell carcinoma was 73.7% in Arm A vs 82% in Arm B as indicated in table 1 patient characteristics and demographics. We included this information both in the Results and the Discussion sections.

19. This places the findings of this small translational study in perspective with the results of KEYNOTE A18, which evaluated concurrent and adjuvant pembrolizumab. Notably, the largest benefit in KEYNOTE A18 was observed in stage III-IVa disease, with the majority of patients (90%) being PDL1 positive and limited benefit observed for adenocarcinomas. The NRG-GY017 study could serve as a foundation for exploring the potential of neoadjuvant immunotherapy in PDL1-negative, locally advanced adenocarcinomas, for which treatment options are currently limited.

19. Response:

Thank you so much for these insightful comments. We have added this sentence to the discussion after highlighting KEYNOTE A18:

“Notably, the largest benefit in KEYNOTE A18 was observed in stage III-IVa disease, with the majority of patients (90%) being PDL1 positive and limited benefit observed for adenocarcinomas. Neoadjuvant sequencing as seen in this trial could serve as a foundation for exploring the potential of neoadjuvant immunotherapy in PDL1-negative, locally advanced adenocarcinomas, for which treatment options are currently limited.”

Reviewer #4 (Remarks to the Author): with expertise in cervical cancer, (immuno)therapy

Jyoti Mayadev and colleagues conducted a phase I trial evaluating neoadjuvant or concurrent Atezolizumab with chemoradiation for locally advanced cervical cancer with 40 patients enrolled. The discussion on optimal sequencing of CRT and ICB is an interesting topic. However, the issues below should be properly addressed

Major concerns

20. Due to the fact that cervical cancer is a chemo-sensitive tumor, there is marginal space for increasing response rates.

20. Response:

Thank you and we appreciate the comment. Unfortunately, while small cervical tumors are indeed curable with chemoradiation alone, this is not the case for high-risk locally advanced cancers, particularly those with nodal involvement, such as the ones explored in our study. Large, randomized trials such as the CALLA trial (Lancet Oncology Dec 2023) and the KEYNOTE A18 trial (Lancet April 2024) continue to show that at 2 yrs. high-risk patients with cervical cancer have rates of recurrence of 33-43%, even with addition of immunotherapy. Thus, in this population, we believe that exploration of strategies to increase cure rates remains an unmet need. We have added this to the Introduction.

21. The expansion of tumor-associated TCR (as a surrogate for anti-tumor immune response) in this study did not seem to translate into prognosis advantage.

21. Response:

We thank you for this comment. Since the number of DFS events in the patients with complete available TCR data was small (n=3 in Arm A and n=5 in Arm B) the study is severely underpowered to draw conclusions about the prognostic significance of tumor-associated TCR expansion. Nevertheless, we have now included this information on a per Arm analysis (Supplementary Figure 5C) and highlighted the limitations in the discussion.

22. Furthermore, the TCR expansion advantage did not persist upon day 21. Please comment.

22. Response:

The reviewer is correct that despite the initial tumor-associated TCR expansion in Arm A (Fig 3E), there was a contraction of these clones in response to radiation. Despite this, on day 21 there was, in fact, a maintained tumor-associated TCR clonal expansion in Arm A when compared to Arm B (Figure 3F, $p=0.052$ by the protocol-specified 2-sided T test). As specified in the methods, this small study was powered to detect an effect size of 0.95 at 10% significance level. Recognizing the potential impact of small sample size, for full transparency in the manuscript we have also included the p-value using non-parametric comparison (Wilcoxon rank sum), which did not meet criteria for statistical significance ($p=0.13$). Nevertheless, while the small size of the study precluded high-power comparisons, we believe that these results support the underlying biologic hypothesis of the study. We provided some clarification for this in the manuscript.

23. The manuscript lacks a clear definition of "tumor-associated clone." Given the importance of this concept to the study's objectives, it is essential to provide a precise definition to ensure basic understanding. Please define the term.

23. Response:

We thank the reviewer for this very important comment. We have provided some clarity to the term in the methods and throughout the manuscript and included an additional figure (Fig. 3b).

24. According to the patient demographics and characteristics, patients in Arm A demonstrated significantly higher pre-treatment PD-L1 (SP263) tumor cell score compared with that of Arm B. Could it be important confounding factor in results interpretation? Please elaborate on this.

24. Response:

We thank the reviewer for the comment but would like to point out that the patients in Arm A actually exhibited a significantly lower pre-treatment PD-L1 tumor cell score. In addition,

Arm A exhibited several other potentially worse clinical factors, such as presence of positive para-aortic lymph nodes. Despite this imbalance, Arm A experienced a higher 2-year DFS and increased immune response. We thus don't believe that these factors served as confounders in results interpretation; in fact, it appears that Arm A had worse clinical and pathological features. In other words, the odds were stacked against Arm A in this study, and despite this, Arm A performed better.

25. Line 117, explain IMRT

25. Response:

We have corrected this to “with Intensity Modulated Radiation Therapy (IMRT)”

26. Line 148 Typo, “randomized”

26. Response:

We have fixed this typo

27. 1. Line 184 Typo, “parametersinclude”

27. Response:

We have corrected this typo

REVIEWERS' COMMENTS

Reviewer #3 (Remarks to the Author):

The authors replied to all my questions. I have no further comments

Reviewer #4 (Remarks to the Author):

The response did not integrate the updated results and figures, which has caused considerable confusion for the reviewers.

Several statistics consider the small sample size; it would be best to increase the sample size rather than just providing explanations.

In particular, it is necessary to discuss the impact of radiotherapy on the activation and expansion of TCR expansion.

The definition of tumor-associated clones is not objective.

REVIEWERS' COMMENTS

Reviewer #3 (Remarks to the Author):

The authors replied to all my questions. I have no further comments

Response: Thank you for the review.

Reviewer #4

Question: The response did not integrate the updated results and figures, which has caused considerable confusion for the reviewers.

Response: We apologize for the confusion. We previously included the summary of our modifications and referenced the new figures in the response letter hoping that it would make it easier for the reader.

Question: Several statistics consider the small sample size; it would be best to increase the sample size rather than just providing explanations.

Response: We completely agree with the small sample size being a limitation. Unfortunately, since this was a prospective randomized trial, we are limited in our analyses to the number of patients that were already enrolled to the trial and the number of biospecimens that we were able to collect. Based on the results of this study a larger clinical trial is currently in development.

Question: In particular, it is necessary to discuss the impact of radiotherapy on the activation and expansion of TCR expansion.

Response: Thank you for this comment and this is indeed one of the most important findings of the studies to emphasize. We have included this in the Results and the Discussion, with focus on all TCR clones and on tumor-associated TCR clones:

Results (line 334): “Overall expansion and appearance of new TCR clones (clonal expansion) from baseline to day 21 was observed in both Arm A ($p < 0.0001$) and Arm B ($p = 0.0002$), though the difference between the arms was not significant ($p = 0.36$) by 2-sided t-test (Figure 3c). Most of the clonal expansion occurred during CRT as evidenced by the clonal expansion after the first cycle in each arm, with significantly higher increase observed in Arm B vs. Arm A (median 132 vs. 76.5, respectively, $p = 0.017$) (Figure 3d). Since overall clonal expansion could be predominantly indicative of a non-specific inflammatory response to radiation, we evaluated the relative expansion of tumor-associated T cell clones out of total expanded clones as a surrogate measure of T cell repertoire that is presumed to be enriched for tumor-reactive T cells (Figure 3a). After cycle 1 of therapy, significantly higher proportion of peripherally expanded clones were tumor-associated in Arm A vs. Arm B (mean 0.37 vs. 0.09, $p = 0.0025$ by a Wilcoxon rank sum test) (Figure 3e).

Results (line 348): “Notably, contraction of the initially expanded tumor-associated clones in Arm A was observed during the CRT phase ($p = 0.048$) (Figure 3g).”

Discussion (line 405): “We find that there was a marked TCR clonal expansion in both arms, which was predominantly observed during the CRT phase. “

Discussion (line 418): Notably these results indicate that most tumor-associated clonal expansion occurred in response to atezolizumab prior to initiation of CRT, suggesting that neoadjuvant immunotherapy may mount an early systemic immune response that is sustained throughout CRT. Finally, while most of the diversity loss during the CRT was likely a nonspecific consequence of CRT-induced leukopenia, we also observed relative reduction of tumor-associated TCR clones in peripheral blood (out of total clones) in response to CRT, particularly in Arm A where these clones have previously expanded (Figure 3g). This would suggest that radiation could indeed have direct negative impact on tumor-associated TCR clones, possibly through their direct killing in the tumor-draining lymph nodes.

Question: The definition of tumor-associated clones is not objective.

Response: We thank the reviewer for this very important comment. We have provided additional clarity to the term in the Methods and included an additional figure (Fig. 3a) for graphic representation.